# Discovering global minimum of High-Dimensional Energy Landscapes

## Abstract

Identifying global minima of high-dimensional non-convex functions is a fundamental problem in fields such as structural biology and materials modeling. Existing solutions (e.g. AlphaFold) often rely on generalizing from data. In contrast, we address the challenging domain where no existing data is available, and only the ground-truth energy function is provided. Utilizing the action functional, we formulate a novel loss function that transforms the input's rough loss landscape into a benign one for the neural network parameters. This allows minimizing the loss to align with finding the global minimum of the energy landscape. We validate our method on high-dimensional global optimization tasks, demonstrating its ability to approximate global minima for energy landscapes with thousand-dimensional inputs.

## 1 Introduction

Identifying ground-truth structures for biomolecules as well as atomic and molecular clusters is essential for fields as diverse as drug discovery, materials science, and catalysis.

Significant progress has been made with regards to the former with AlphaFold3. However, 75% of its training data relies exclusively on proteins. Thus, it lacks accuracy with regards to other types of biomolecules. For instance, it is not able to predict the RNA structure for RNAs from orphan families (Zonta & Pantano, 2024).

For atomic and molecular clusters, the situation is even more dire. Training data is sparse, precluding the development of an AlphaFold-like model.

For both of these problems, additional ground-truth structures must be acquired experimentally, a laborious process. However, we have access to ground-truth energy functions that are cheap to evaluate. For biomolecules, the AMBER force field can accurately model the energy of biomolecules (He et al., 2020). With regards to atomic and molecular clusters, the ground-truth structure can be found by minimizing a combination of lennard jones and coloumb potentials. This ground-truth structure, typically, is then used to warm-start a density functional theory calculation (Marques et al., 2017).

Instead of generalizing from existing training data, the holy grail would be to utilize the accurate, cheap-to-query energy function to find the global minimum. This would sidestep the siloed accuracy of existing models. This is the focus of the paper.

## 2 Related Work

There are a total of three papers that attempt to tackle this problem via deep learning.

Firstly, Bihani et al. (2023) minimizes energy functions via reinforcement learning applied to graph neural networks. They are able to find points that match or slightly improve on the lowest energy points found by Adam. However, they do not include these points' distance from the global minimum.

We reproduce one of their experiments in Appendix A.4, and find that their method does not produce points that are closer to the global minimum than Adam. This is not surprising as the energy

values of a configuration are not correlated with the point's distance from the global minimum. This can be visualized on a small-scale by examining the dis-connectivity graph of the 38-dimensional lennard jones function in Cameron (2014). There is significant variation in the energies of the local minima. However, these energies are not correlated with the local minima's distance from the global minimum. The Adam optimizer, for instance, is able to substantially lower the energy value. However, its final configuration's distance from the global minimum is not much better than randomly sampling a point.

The second paper is Merchant et al. (2021), which utilizes a learned optimizer to find the global minimum. However, due to rough loss landscapes with respect to their neural network parameters, their model can't even be trained via gradient descent. Instead, meta-gradients are estimated via parameter perturbations. Thus, their method cannot find the global minimum of a problem with more than 70 dimensions.

The third paper is Zhao et al. (2022), which construct a new class of neural networks from lattice functions, which are interpolated look-up tables. Their network's unique global minimizer can be extracted efficiently. However, due to limitations in expressivity, they demonstrate results on at most 61 dimensions.

Thus, while this is a central problem, research remains limited. Furthermore, existing approaches do not appear to make progress in identifying the global minimum for functions beyond 100 dimensions.

## 3 PROBLEM FORMULATION

The first step is to understand the nature of the problem. Are neural networks able to identify the global minimum, and is it simply a problem of extraction? Or are neural networks, more fundamentally, not able to identify the global minimum?

Zhao et al. (2022) indicates the problem is the latter. They extensively document how neural networks are not able to identify the global minimum for toy problems. For instance, on the 4-dimensional Rosenbrock function, a deep neural network trained with 10,000 points is not able to identify the minimal point in the test set.

Under the Neural Tangent Kernel assumption, neural networks behave like kernel regression models, interpolating between training points. As a result, these models are not well-suited to extrapolating beyond the range of the training data. Thus, they are not well-suited to identifying global minimum.

This reveals a deeper issue: the minimal y-value of a global minimum is not, by itself, sufficient for its identification. Therefore, we need to exploit additional properties of the global minimum that neural networks can leverage.

## 4 METHODS

### 4.1 ACTION FUNCTIONAL LOSS

We can model the biomolecule's trajectory with the overdamped Langevin equation, where $U$ is the energy function of interest, $C$ is a constant, and $dW_t$ is a Wiener process.

$$dX_t = -\nabla U(X_t)\,dt + \sqrt{C}\,dW_t$$

The only assumptions here are that inertia is negligible and noise is independent of state, which are standard assumptions for biomolecule trajectories.

From this stochastic differential equation, we can characterize the probability of transitioning from one state to another state (Adib, 2008).

$$P(x_f \mid x_i; t) = \int_{x(0)=x_i}^{x(t)=x_f} \mathcal{D}x(s)\, e^{-S[x(s)]/C} \tag{1}$$

Here, $x_i$ and $x_f$ denote the initial and final states of the protein, respectively. $x(s)$ is a trajectory that begins at $x_i$ and ends at $x_f$. Furthermore, the action, $S[x(s)]$, is proportional to the trajectory's mean squared error deviation from gradient descent. The expression integrates over all possible trajectories with the fixed starting and ending states.

Thus, we can express the probability of transitioning from random initial points to a specified final point as below.

$$P(x_f) = \int_{x_i \sim \mathrm{Uniform}([-B,B]^D)} P(x_f \mid x_i; t)\, dx_i \tag{2}$$

$P(x_f)$ directly captures our quantity of interest; it's proportional to the amount of time a folded protein would spend in conformation $x_f$.

The $x_f$ that maximizes this expression is the most-likely ground state, or the global minimum.

### 4.2 PRACTICAL IMPLEMENTATION

We aim to minimize $P(x_f)$ in the simplest manner possible. $P(x_f)$ has four components: uniform sampling of states $x_i$, integral over all paths $x(s)$ between $x_i$ and $x_f$, identifying a path $x(s)$, and calculating the action for a given path.

The first component, uniform sampling of states $x_i$, is straightforward. We identify how to estimate the remaining three components in the three sections below.

#### 4.2.1 APPROXIMATION FOR PATH INTEGRAL

The above expression integrates over all paths between $x_i$ and $x_f$. Since the action $S[x(s)]$ is weighted exponentially, in the limit of low temperature or high friction, the path integral is dominated by the most probable path. Thus, we approximate this integral with the most probable path between $x_i$ and $x_f$.

#### 4.2.2 FINDING PATH $x(s)$ BETWEEN $x_i$ AND $x_f$

How do we derive the most probable path mentioned in the previous section?

For a given $x_f$, we represent the optimal paths between any $x_i$ to the $x_f$ via a neural network that contains no local minima, by construction. The $x_f$ is the global minimum of the neural network.

Therefore, from a starting $x_i$, one can perform gradient descent on the function learned by the neural network to get the trajectory to $x_f$.

The neural network contains no local minima by construction because it is a composition of an invertible network and a strictly convex function.

#### 4.2.3 APPROXIMATION FOR ACTION

The action for a trajectory is proportional to the trajectory's mean squared error deviation from gradient descent. However, we instead penalize the mean squared error loss between the neural network's predicted y-values for the trajectory and the ground-truth energy values for the trajectory. While this is not equivalent to penalizing the neural network's gradient, we find that it works well in practice.

### 4.3 TRAINING SETUP TO MINIMIZE $P(x_f)$

We utilize the three approximations above to construct a training scheme that minimizes $P(x_f)$.

First, we uniformly sample our initial starting points, $x_i$.

Next, we find the most probable path $x(s)$ between $x_i$ and the neural network's learned global minimum, $x_f$, by applying gradient descent steps to $x_i$ until we reach $x_f$. This works because, as

mentioned above, the neural network has no local minima by construction. This is in accordance with the approximations from Section 4.2.1 and Section 4.2.2.

Thirdly, we evaluate the action $S[x(s)]$ along the trajectory $x(s)$ by measuring the mean squared loss between the ground-truth energy values for the trajectory, $U(x(s))$, and the neural network's predictions for the trajectory, $f_\theta(x(s))$. This is in accordance with the approximation from Section 4.2.3.

Lastly, we sum this loss across all trajectories to estimate the integral in Equation 2.

We update the neural network's parameters to minimize this loss. Given that this loss is an estimate of $P(x_f)$, our training scheme minimizes $P(x_f)$ over time.

Furthermore, we warm-start the optimization process by training the neural network to mimic $U(x)$ for randomly sampled $x$.

The above process is outlined in Algorithm 1 and illustrated in Figure 1.

---

**Algorithm 1** Training Setup for Action Functional Loss Minimization

---

1: **Input:** Neural network $f_\theta(x)$, Ground-truth energy $U(x)$, bound $B$, learning rate $\eta$, epochs $N$, global minimum threshold $\epsilon$
2: **Note:** $f_\theta(x)$ has no local minima by construction.
3: **for** epoch $= 1$ to $N$ **do**             ▷ Learning $U(x)$ through random samples
4:     **for** $x_i \sim \text{Uniform}([-B, B]^D)$ **do**
5:         $\hat{y}_i = f_\theta(x_i)$
6:         $y_i = U(x_i)$
7:         $L = \frac{1}{n}\sum_i(\hat{y}_i - y_i)^2$
8:         $\theta \leftarrow \theta - \nabla_\theta L$
9:     **end for**
10: **end for**
11: **for** epoch $= N + 1$ to $2N$ **do**          ▷ Minimizing Action Functional Loss
12:     $L = 0$
13:     **for** $x_i \sim \text{Uniform}([-B, B]^D)$ **do**        ▷ Estimating Integral in Equation 2
14:         trajectories = []
15:         **while** max pairwise distance between $x_i$ is $\geq \epsilon$ **do**      ▷ Pending convergence
16:             $x_i \leftarrow x_i - \eta\nabla_x f_\theta(x_i)$
17:             trajectories.append($x_i$)        ▷ Estimating $x(s)$ from Section 4.2.2
18:         **end while**
19:         $\hat{y}_i = f_\theta(\text{trajectories})$
20:         $y_i = U(\text{trajectories})$
21:         $L_i = \frac{1}{n}\sum_i(\hat{y}_i - y_i)^2$        ▷ Estimating $S[x(s)]$ from Section 4.2.3
22:         $L = L + L_i$
23:     **end for**
24:     $\theta \leftarrow \theta - \nabla_\theta L$
25: **end for**

---

## 5 EXPERIMENTAL SETUP

We test our method on two global optimization tasks; minimizing the Alpine function and the Schaffer F7 function. Both functions have a global minimum at 0. To ensure the task is challenging, we minimize over a thousand dimensions, with each dimension ranging from [-100,100].

For baselines, we utilize the Adam optimizer (Kingma & Ba, 2015) as well as only training the neural network on random samples (e.g. omitting the Minimizing Action Functional Loss component in Algorithm 1).

### 5.1 MODEL ARCHITECTURE

For the architecture to only have one minimum, we need an invertible network followed by a strictly convex function. We adopt the PLNet architecture developed in Wang et al. (2024). We use an

$$P(x_f) = \int_{x_i \sim \text{Uniform}([-B,B]^D)} \left( \int_{x(0)=x_i}^{x(t)=x_f} \mathcal{D}x(s)\, e^{-S[x(s)]/C} \right) dx_i$$

Approximated by
120 samples

Approximated
by trajectory to
learned global
minimum

Approximated by
$(f_\theta(x(s)) - U(x(s)))^2$

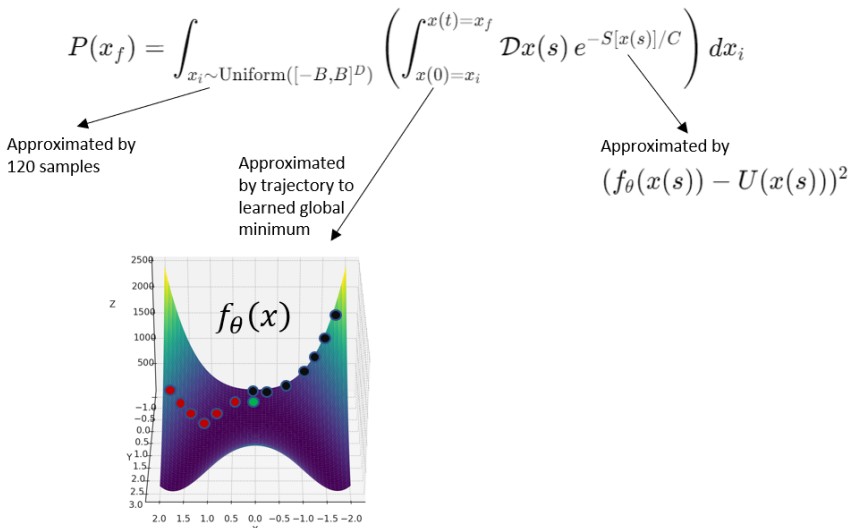

Figure 1: Summary of method

architecture with four blocks, where each block has 8 layers, each with 1024 hidden units. $\tau$, which controls the Lipschitz constant, is set to be a learnable parameter.

## 5.2 TRAINING

For the first 200 epochs, the model was trained to mimic the ground-truth energy function on 1000-dimensional points that were uniformly sampled from $[-100, 100]^{1000}$. The y-value was scaled such that the mean and standard deviation of the uniformly sampled points were 0 and 1, respectively.

Afterwards, we sampled 120 random initial points, from the same distribution as above. Using the Adam optimizer, we perform gradient descent on these points with respect to the model. Eventually, we reach the unique global minimum as the model, by construction, does not have local minima.

We then filter these trajectories so that they are evenly spaced. We calculate the ground-truth energy value for the filtered trajectories, and use this as the model's training data for the next epoch.

Every epoch, we calculate the model's global minimum and output the global minimum with the lowest energy as the final prediction for the global minimum.

## 5.3 TEST FUNCTIONS

Plevris & Solorzano (2022) catalog 30 functions for a global optimization benchmark. Out of these 30, we identify the functions that have more than one minima, have a defined global minimum within $R^d$, and are continuous and differentiable. This narrows the scope to 14 functions.

Out of these 14 functions, we see if an Adam optimizer can find the global minimum within a distance of 5, starting from thousand-dimensional points uniformly sampled from $[-100, 100]^{1000}$. This narrows the scope to 8 functions. This step is necessary as we only want to test on challenging functions that Adam cannot handle.

Next, we test if a fully connected neural network with 5 layers, with 1000 hidden units each, trained on randomly sampled points from $[-100, 100]^{100}$ can model the function more accurately than using the mean of the y-values of the batch as the prediction. We would like to test if the function is "learnable" by a neural network, in the most basic sense. Note that this is an unconstrained, standard fully connected neural network. This narrows the scope to two functions: Schaffer's F7 and Alpine 1.

The functions in the benchmark are manually engineered to contain high frequency components with large magnitudes. Abbe et al. (2023) characterizes the types of high-frequency functions that

Table 1: Results for optimizing schaffer f7 function with 1000-D inputs sampled from uniform($[-100, 100]^{1000}$), with each method run once with seed 0.

| Method | Min Dist From Global Min | Min Dist From Global Min For Min Energy | Avg Error Per Dim |
|---|---|---|---|
| Our Method | 102.4842 | 102.4842 | 3.2408 |
| Standard Baseline | 495.8499 | 495.8499 | 15.6802 |
| Adam Baseline | 1744.2198 | 1701.7724 | 55.1571 |

cannot be efficiently learned by neural networks. However, biological functions of interest have been well-modelled by neural networks. Thus, while the above step of requiring the function to be "learnable" by a neural network eliminated 6 functions in the benchmark, we don't consider it to be limiting in practice.

Further details can be found in the Appendix A.1.

Both Alpine 1 and Schaffer's F7 functions are highly non-convex with many local minima. The global minimum of these functions is 0. As mentioned above, the inputs are 1000 dimensional and each dimension can range from [-100,100].The equations for both functions are below.

### 5.3.1 ALPINE 1 FUNCTION

$$f(\mathbf{x}) = \sum_{i=1}^{n} |x_i \sin(x_i) + 0.1 x_i|$$

### 5.3.2 SCHAFFER'S F7 FUNCTION

$$f(\mathbf{x}) = \frac{1}{n} \sum_{i=1}^{n-1} \left( (x_i^2 + x_{i+1}^2)^{0.25} \cdot \left( 1 + \sin^2 \left( 50 \cdot (x_i^2 + x_{i+1}^2)^{0.1} \right) \right) \right)$$

### 5.4 BASELINES

We use two baselines. The first is gradient descent using the Adam optimizer. We select the point with the lowest energy as the final prediction. Further details can be found in Appendix A.3

The second baseline is to only train the model to mimic the ground-truth for points uniformly sampled from $[-100, 100]^{1000}$. This is the same training scheme used for the first 200 epochs in our method, but for the baseline, we continue this indefinitely. Thus, we are able to assess whether the improvement is due to our method or simply due to additional training data. Similar to our method, we calculate the model's learned global minimum each iteration and output the global minimum with the lowest energy as the final prediction for the global minimum. This baseline and our method were each run for 20 hours on a V100 GPU.

## 6 RESULTS

From Table 1 and Table 2, we can see that the proposed method finds the global minimum with an average of 3-4% error for each dimension. This is in contrast to the baseline, which averages an error of 15-17% per a dimension. Adam is even worse, with its error ranging from 32-55%.

Training a neural network without local minima to approximate the function offers a significant improvement over Adam. However, it is not enough in and of itself. For the neural network to precisely identify the global minimum, our method's training scheme is needed.

In Figure 2 and Figure 3, we can see that quality of the global minimum learned by the baseline does not improve over time. This is in marked contrast to our method, which shows the learned global minimum improving rapidly in the first several iterations.

Table 2: Results for optimizing alpine 1 function with 1000-D inputs sampled from uniform($[-100, 100]^{1000}$), with each method run once with seed 0.

| Method | Min Dist From Global Min | Min Dist From Global Min For Min Energy | Avg Error Per Dim |
|---|---|---|---|
| Our Method | 113.8799 | 113.8799 | 3.601 |
| Standard Baseline | 533.1343 | 545.4366 | 17.2482 |
| Adam Baseline | 482.7661 | 1006.0282 | 31.8134 |

### 6.0.1 ALPINE 1 FUNCTION

In Figure 2, we calculate the learned global minimum each iteration. We then measure the quality of this minimum by calculating its energy in addition to its distance from the true global minimum, 0. Under the baselines, in red, we can see that the quality of the learned global minimum under the baseline method remains stagnant. Under our method, in blue, the quality of the learned global minimum rapidly improves. The best learned global minimum is found after 14 iterations.

Furthermore, for our method, we see an almost perfect correlation between the energy and distance from the true global minimum. This is not the case with the baseline.

Lastly, after 14 iterations, our method has a marked divergence, where the quality of the learned global minimum gets worse. This is due to our action approximation described in Section 4.2.3. The true action is proportional to the trajectory's deviation from gradient descent. However, we approximate this by measuring the loss for the y-values along the trajectory.

We see that the neural network is able to "game" the loss by producing trajectories that deviate from gradient descent. However, it is able to achieve low training loss on these trajectories by accurately modelling the energy values. This only happens in later iterations as the model deviates further from $U(x)$, the ground-truth energy function it was trained to mimic for the first 200 epochs.

Furthermore, we can only model $U(x)$ with a neural network, as opposed to $U'(x)$. This is because, for the latter, we cannot create an architecture, such that by construction, gradient descent trajectories from different initial conditions converge to a unique fixed point.

There are a number of ways to correct this issue (penalizing the model's gradient to mimic the gradient of the true energy function, continually training on random samples from $U(x)$, etc.). We have not yet figured out the optimal fix. However, the existing method is able to produce satisfactory results as this only becomes a problem in later iterations. Given that evaluating the energy function is extremely cheap, we are able to simply select, across iterations, the learned global minimum with the lowest energy.

We briefly explain why the number of iterations are different for both methods. Both experiments were run for 20 hours on a GPU. However, since our method dynamically generates its training data for the next epoch via its gradient descent trajectories, it has more data points per an epoch than the baseline model. Thus, it's able to finish fewer iterations during the same amount of time.

### 6.0.2 SCHAFFER'S F7 FUNCTION

We see the same patterns as the Alpine function optimization. The baseline remains stagnant whereas our method converges quickly and then diverges.

See above for explanations for the divergence and the differing number of iterations between experiments.

### 6.0.3 TRAJECTORY LOSS

Below, we look at the trajectory loss over time for both functions. This can be seen in Figure 4. The initial spike in the loss is due to the model adapting to its rapidly changing training distribution. After this, the loss decreases steadily. This indicates that loss landscape for the neural network's parameters is well-behaved.

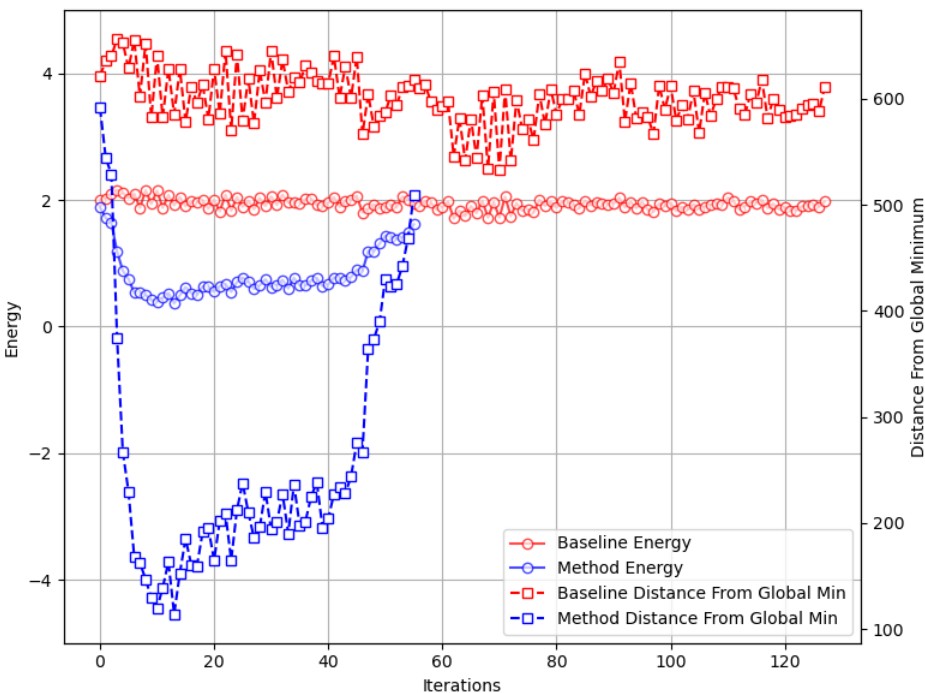

Figure 2: Results from alpine function optimization. We display the energy and distance from the true global minimum for the learned global minimum over iterations. For the baseline, we simply train the neural network with randomly sampled points. Thus, the quality of its learned global minimum does not improve over time. By comparison, the quality of the learned global minimum under our method rapidly improves, converging in the first 14 iterations.

While the loss is decreasing, the trajectory loss does not tightly correspond to $P(x_f)$ from Section 4.1, due to the approximation for calculating the action outlined Section 4.2.3.

## 7 DISCUSSION

With our method, we are able to identify the global minimum for high-dimensional (1000 dimensions) tasks, with a per-dimension error 3-4%. This is not possible to do via other proposed methods such as reinforcement learning (RL) or learned optimizers.

There are two reasons why this is the case.

Firstly, RL and learned optimizers upweight states based on their energy values. However, the energy values and gradients of a configuration are largely determined by the configuration's distance to its local minima. Thus, this is not a useful signal. By contrast, our method *directly* minimizes $P(x_f)$ from Equation 2, with respect to the neural network's parameters $\theta$, as opposed to $x$. Thus, we are able to sidestep the dominance of the local minima.

Secondly, our method uses standard supervised learning. This is in contrast to RL and learned optimizers, which suffer from brittle loss landscapes, extreme hyperparameter sensitivity, and slow training times due to their unorthodox training schemes.

Due to these two innovations, our method enjoys the smooth, well-behaved loss landscapes that are typical of neural network training. This can be seen in Figure 4, where the trajectory loss of our method steadily decreases.

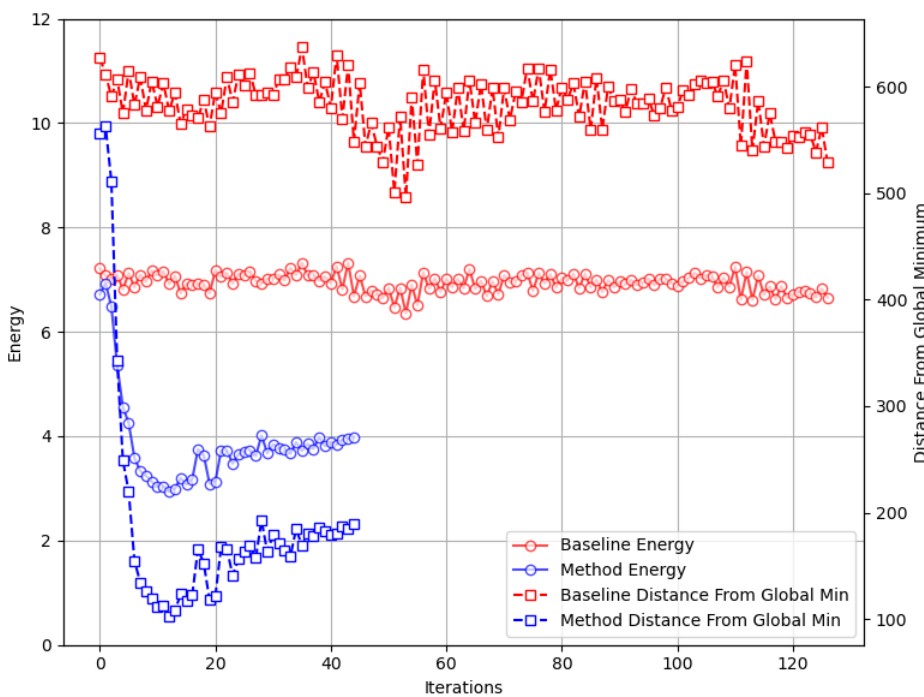

Figure 3: Results from schaffer function optimization. We display the energy and distance from the true global minimum for the learned global minimum over iterations. For the baseline, we simply train the neural network with randomly sampled points. Thus, the quality of its learned global minimum does not improve over time. By comparison, the quality of the learned global minimum under our method rapidly improves, converging in the first 13 iterations.

## 7.1 LEARNING SIGNAL

### 7.1.1 ENERGY VALUES ARE A POOR LEARNING SIGNAL

Both RL and learned optimizers learn by prioritizing states with low energy values. However, as discussed above, a lower energy value has close to no relationship with whether a configuration is closer to the global optima. The number of local minima increases exponentially with dimension, and the function is dominated by high-frequency components with large magnitudes. Thus, the majority of signal in the gradient and variation in the energy values is driven by proximity to local minima. However, proximity to local minima is nearly useless for identifying the global minimum. Thus, the gradient update is not meaningful.

A configuration's energy value only becomes correlated with the signal of interest, the configuration's distance from the global minimum, for configurations that are *already* close to the global minimum. This can be visualized in the dis-connectivity graph of a 784-dimension lennard jones function (de Souza & Wales, 2016).

### 7.1.2 OUR METHOD'S LEARNING SIGNAL

However, our method, by construction, directly minimizes $P(x_f)$ from Equation 2. Note that this minimization is *direct*. After the neural network's parameters $\theta$ undergo a step of gradient descent, the neural network function, $f_\theta(x)$ more closely resembles the ground-truth energy function, $U(x)$. In contrast, a step of gradient descent for neural network parameters trained via learned optimization or reinforcement learning, only trains $f_\theta(x)$ to identify local minima.

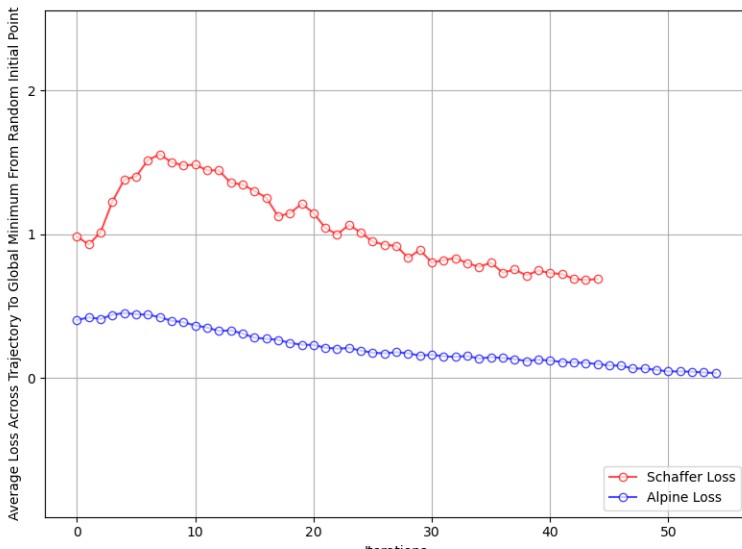

Figure 4: Average Y Loss on Gradient Descent Trajectories Over Time for Our Method. We evaluate the accuracy of the neural network's predicted y-values on the gradient descent trajectories. After an initial spike to adjust to a rapidly shifting training distribution, this loss decreases over time, indicating a favorable optimization landscape.

Thus, we have transformed an optimization problem with respect to $x$, which has a rough energy landscape, into an optimization problem with respect to the neural network's parameters $\theta$, which has a smooth landscape due to the high-dimensionality of $\theta$.

### 7.2 UTILIZING SUPERVISED LEARNING

Even in other domains where the reward is representative of the goal, RL and learned optimizers are notoriously difficult to train due to their high variance and reliance on back-propagating through long trajectories.

By contrast, our method utilizes standard supervised learning. Therefore, we are able to inherit its benign loss landscapes.

## 8 CONCLUSION

Our paper is the first to identify global minima for high-dimensional, non-convex optimization problems. We identify global minima for 1000-D problems with a 3-4% per-dimension error rate. We are able to do so via exploiting a unique feature of biological global minima: regardless of initial state, there exists a path (with minor deviation from mean squared error) that connects the initial state to the global minima. Our method avoids the scalability problems of previous approaches due to two properties. Firstly, minimizing the loss with respect to the model parameters corresponds to recovering the global minimum. Thus, we avoid the rough loss landscapes and proliferation of local minima that arise when minimizing the energy function with respect to the input, $x$. Secondly, our method harnesses standard supervised learning, inheriting its ease of optimization.

## 9 REPRODUCIBILITY STATEMENT

The supplementary zip file contains the code to reproduce the entirety of the experimental section. The purpose of each file is described in the README.

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

## A  Appendix

### A.1  Selecting Test Functions to Benchmark On

From the benchmark of 30 global optimization functions (Plevris & Solorzano, 2022), we identify the following 14 functions as having more than one minima, a defined global minimum within $R^d$, and being continuous and differentiable. These 14 functions are drop-wave, alpine 1, ackley, griewank, rastrigin, happycat, hgbat, rosenbrock, schaffer's f7 function, expanded schaffer's f6 function, salomon's function, xin-she yang 1, modified xin-she yang 3 function, and modified xin-she yang 5 function.

For the second filtering step, we find 8 functions where Adam is not able to find a point within a distance of 5 from the global minimum. Given that our inputs are 1000 dimensional and range from [-100,100], a distance of 5 from the global minimum means the average coordinate norm is 0.16, indicating an average error of 0.16%. Our hyperparameters for testing Adam are described in the Section A.3. These 8 functions are alpine 1, rastrigin, schaffer's f7 function, expanded schaffer f6 function, xin-she yang 1, modified xin-she yang 3 function, modified xin-she yang 5 function, and ackley function.

For the third filtering step, we see if the remaining 8 functions are learnable by a standard, unconstrained fully-connected neural network. To test this, we train a fully connected neural network with 5 layers and 1000 hidden units for 1000 steps and a learning rate of 0.001. For each step, we uniformly sample 1000 points from $[-100, 100]^{100}$. The data is normalized a priori so that the mean and standard deviations of the y-value are 0 and 1, respectively. We train the neural network on 100 dimensional points, despite running our experiments on 1000 dimensional points, for computational efficiency. We consider the function "learnable" if the final loss from the neural network is lower than the loss from simply imputing the mean of the y-value of the batch. Note that this is a very liberal definition of "learnable". Out of the 8 remaining functions, we find that only two are "learnable". These two are Alpine 1 and Schaffer's F7 functions. Thus, these are the two test functions we perform experiments on.

### A.2  Training Details

The gradient norm is clipped at 0.5 throughout training. The model was not able to learn from the sampled trajectories without gradient clipping. Each experiment was run for 20 hours on a V100 GPU, with the random seed fixed at 0.

### A.3  Benchmarking Against Adam

We run 10,000 steps of gradient descent for 10,000 points uniformly sampled from $[-100, 100]^{1000}$. We run the above for every learning rate from 0.05 to 5.0, with increments of 0.05. Across all of these trajectories, we select the points with the lowest energy as the final prediction for the global minimum.

### A.4  Benchmarking Against StriderNET

The experiments described in  Bihani et al. (2023) do not have known global minima. Thus, we adapt the $\sigma$ and $\epsilon$ values of their 100-D binary lennard jones experiment to match an experiment appearing in  Mravlak et al. (2016), a catalog of global optima for binary lennard jones across different hyperparameters. Specifically, we change $\sigma_{AB}$ and $\sigma_{BB}$ to be 1 from 0.8 and 0.88, respectively. Furthermore, we change $\epsilon_{BB}$ from 0.5 to 0.62. We keep all other parameters, including the number of particles, 100, and the A:B proportions, 80:20, to be the same.

We run the code provided on the github repository associated with StriderNET, without changing any of the default hyper-parameters, including the radius cutoffs. Furthermore, we utilize the DatasetLJ numpy file provided in the repo as the initial conditions for the simulation. We take the configuration with the lowest energy throughout the simulation as the final prediction.

We find that this configuration has a distance of 28.79 from the global minimum. Unlike  Bihani et al. (2023), we found that gradient descent with an Adam optimizer outperformed StriderNET

in producing lower energy configurations. We found that the lowest energy point from the Adam optimizer when the binary lennard jones function adhered to StriderNET's cutoff hyper-parameters had a distance of 29.90 from the global minimum. Furthermore, the lowest energy point from the Adam optimizer for minimizing the binary lennard jones function, without any radius cutoff applied, had a distance of 26.34.

