# OpenReview forum: "Discovering Global Minima of High-Dimensional Energy Landscapes"
_ICLR.cc/2025/Conference — ICLR 2025 Conference Withdrawn Submission_

### Official Review · Reviewer_dK44 · 2024-11-03

**Soundness:** 2
**Presentation:** 2
**Contribution:** 2
**Rating:** 3
**Confidence:** 3

**Summary:**

The paper under review identifies an important problem with great potential is molecule generation, analyzes the current obstacles and proposes a solution. The approach suggested is innovative and reasonable and is in my opinion a promising path. However the paper has a couple of substantial drawbacks: (1) unclear formulation of methodology; (2) experiments were only carried out on limited synthetic data and not convincing enough in terms of generalizability.

**Strengths:**

The paper aims to solve the problem of optimizing over a rough landscape, which is a common scenario in finding ground-truth structures for molecules and clusters. This is an important domain where current methods have limited applicability due to lack of training data and requires more fundamental approaches. The authors proposes to mimic the landscape by a class of neural networks that don't have local minima, which is a novel idea. Experiments demonstrate clear advantage of this approach over Adam and StriderNet on two test landscape functions.

**Weaknesses:**

I find room for improvements in two aspects :

1. Unclarity of description. The new method was first formulated in words in section 4.2 and then algorithmically in section 4.3. However both sections contain ambiguities that obstruct the reader's understanding. For example:
-L111. What is t?
-L146-L147 are not very clear, would you mind elaborating more?
-L193-L194. The notation is very confusing here: are the inputs of f_\theta and U points x, or trajectories of points? It seems you are using both (line 5-6 vs line 19-20 in the algorithm) without warning. If f_\theta(x) is well defined, by f_\theta(trajectories) did you mean the min or average over points inside one trajectory, or maybe over all trajectories?
-It was explained that the architecture of f_\theta in implementation was the PLNet in (Wang et al. 2024). I would love to see more details, as in that paper, PLNet refers to a class of neural nets satsifying a mathematical condition, rather than a concrete architecture, and freedom in implementation was mentioned therein.

2. Lack of realisitic experiments
The authors cherry-picked two test functions from a pool of 30. While I believe that these are the only ones from that pool that required a new algorithm for optimize, I feel there is a danger of implicit overfitting here by representing all rough loss function with only two test function, especially as those test functions are synthetic. In particular, one can image landscape functions that have a smooth trunk that is L^2-close to a convex function or PLNet, as well as scattered downward spikes. By the nature of the new algorithm, one may suspect that it can overfit the smooth trunk and ignore the downward spikes. Neither the theoretical arguments nor the experiments from the paper helped to exclude such a possibility.

**Questions:**

My questions are formulated in the weakness section above. I guess the expository ambiguity could be easily improved in revision. I wish the authors could discuss more representativeness of test functions.

---

### Official Review · Reviewer_hjHK · 2024-11-03

**Soundness:** 1
**Presentation:** 2
**Contribution:** 1
**Rating:** 3
**Confidence:** 4

**Summary:**

The manuscript suggests an optimization algorithm that, according to the authors, is able to find the global minimum in general non convex functions in high dimensions. Basically, their method consists on first training a neural network to mimic the target function and then doing gradient descent on the network.

**Strengths:**

If the authors' claim is correct, then they have found the holy grail of a few sceintific fields. Finding the global miminum of a general function is quite a feat. Unfortunately, I do not think the proposed method delivers this.

**Weaknesses:**

The paper is poorly written both in terms of framing the contribution in the right context, and in describing in detail what the contribution is. Specifically:

- The method is presented as a completely general method with no discussion about in which cases it is applicable. I find it highly unlikely that the method is as general as presented, no analytical justification for this is given and the empirical evidence is far from convincing

- The literature review is quite lean. The authors state (line 47) that "There are a total of three papers that attempt to tackle this problem via deep learning.", where by "this problem" they refer to finding the global minimum of a function. Admittedly, I do not know of many papers who claim they have a general method that does this in every context, but there are hundreds of papers who address this problem in a specific context using DL.

- The benchmarks are not SOTA. There are hundreds of optimization methods out there, each tailored for different use cases. Comparing just to GD with ADAM is far from convincing. A few suggestions to compare to: Nelder-Mead, conjugate gradient, BFGS/L-BFGS.

- The algorithm is very poorly presented. see questions section below.

- many statements are vague and ill defined. for example:
  * line 121: "The $x_f$ that minimizes this expression is the most-likely ground state, or the global minimum". what is "most likely"? under which distribution? is there a proof? citation? is "most likely" the same is "global minimum" in the presence of temperature?
  * line 109: "The action for a trajectory is proportional to the trajectory’s mean squared error deviation from gradient descent". what is meant by "squared error deviation from GD"? proper definitions should be used.

**Questions:**

- If the neural net has no local mimina by construction, how can it mimic the ground truth function that has multiple mimina?
- I could not understand what the algorithm does from its description (Algorithm 1):
  * What is a trajectory? from line 17 it seems that each $x_i$ is a trajectory? or is it a list of points? what does it mean to apply f to the trajectory (line 19), if f takes in a single point as input?
  * are the $x_i$ in line 13 randomized at each iteration of the for loop in line 11?
  * what does the index $i$ do in line 21? how come it's summed over in the right hand side but still appears in the left hand side?

---

### Official Review · Reviewer_tovh · 2024-11-04

**Soundness:** 1
**Presentation:** 1
**Contribution:** 1
**Rating:** 1
**Confidence:** 4

**Summary:**

An algorithm to search for ground-truth structure of biomolecules is presented. The method is based on utilizing the action functional. Numerical benchmark on test functions is conducted.

**Strengths:**

This paper appears to be in its very early stages.

**Weaknesses:**

The presentation requires significant improvement, the related work section does not cover enough state-of-the-art approaches, and the model’s explanation remains unclear. Additionally, the numerical results are inconclusive. Given the substantial advancements in this area (such as RoseTTAFold, GOFEE, and AlphaFold 3), this work does not currently present a compelling novelty that would justify acceptance in its current form.

**Questions:**

1) What is a "benign" loss landscape?

2) How do the test functions used for benchmarking related to the target problem of finding biomolecule ground-truth structures?

3) Why is the path integral approach used rather than just simulating the Langevin equations?

4) Why is the PLNet architecture (Wang2024) used here rather than another approach?

---

### Official Review · Reviewer_CfEa · 2024-11-04

**Soundness:** 1
**Presentation:** 1
**Contribution:** 1
**Rating:** 1
**Confidence:** 3

**Summary:**

If I understand the paper, which is very poorly written and very hard to understand, the authors propose a new algorithm for solving global minimization problems in rough landscapes.
The algorithm trains a neural network to model the real potential, in two different phases
that one could term a static one, just sampling points, and a dynamic one, where the gd dynamic through the network is followed. They don't explain why but they state that by construction has a single minimum.

**Strengths:**

There could be something good to the methodology, but this is very hard to understand given the low presentation quality of the paper.

**Weaknesses:**

The paper is very poorly written, the algorithm is poorly motivated, everything is hard to understand,
even the pseudocode they write is unintelligible due to repeated index i. The experimental section is also very poor. This is such a low quality submission that I'm not going to waste my time detailing what is wrong with
the paper.

**Questions:**

I have no questions, since the limitations of the paper cannot be addressed in a rebuttal.

---

### Official Review · Reviewer_ZDtw · 2024-11-05

**Soundness:** 3
**Presentation:** 3
**Contribution:** 2
**Rating:** 5
**Confidence:** 3

**Summary:**

This paper introduces a new approach to addressing rough loss landscapes for global optimization by leveraging a novel loss function inspired by the action functional. While the results are competitive with some neural network-based state-of-the-art methods, adding context with traditional methods from physics and chemistry could broaden its relevance and strengthen the claims. With these additions, the paper would offer a more comprehensive view of the method's advantages and potential applications.

**Strengths:**

(1)  The authors present an new approach by utilizing the action functional to reformulate the loss landscape into a more manageable shape. This idea of transforming a "rough" loss landscape into a "benign" one for optimizing neural network parameters is neat and innovative and could offer many advantages in challenging high-dimensional optimization tasks.

(2) The paper's validation on high-dimensional global optimization tasks is well-executed, demonstrating that the proposed method can indeed approximate global minima in complex energy landscapes. This approach appears to be competitive with, or even comparable to, state-of-the-art neural network approaches.

**Weaknesses:**

(1) The proposed method is based on overdamped Langevin systems, which approximate low-temperature, high-friction scenarios, thereby limiting its applicability in broader settings where inertia plays a significant role.
(2) One significant gap is the lack of comparison with well-established global optimization methods from the physics and chemistry domains. These fields have a rich history of research in energy landscape exploration.

**Questions:**

(1) The paper could benefit from a deeper discussion of the specific challenges associated with high-dimensional spaces. Many physics-based methods address this through various dimensionality reduction techniques or by leveraging symmetries. Also, I would like to see if this paper could consider additional benchmarking against problems that are widely used in both the physics and chemistry communities.

(2) Can the authors of this paper discuss extension of proposed method to more general systems, such as the inertial Langevin systems?

---

### Note · Authors · 2024-11-13

I have read and agree with the venue's withdrawal policy on behalf of myself and my co-authors.